



# Groundwater fluctuations during a debris flow event in Western Norway – triggered by rain and snowmelt

Stein Bondevik[1], Asgeir Sorteberg [2]

[1]Department of Environmental Sciences, Western Norway University of Applied Sciences, P.O. Box 133, NO-6851 Sogndal, Norway
[2]Geophysical Institute, University of Bergen and Bjerknes Centre for Climate Research, University of Bergen, Norway

*Correspondence to*: Stein Bondevik (stein.bondevik@hvl.no)

**Abstract.** Pore pressure is crucial in triggering debris slides and flows. Here we present measurements of ground water pore pressure and temperature recorded by a piezometer 1.6 m below the surface on a slope susceptible to debris flows in Western Norway. One of the largest oscillations in data collected over four years coincided with a debris flow event on the slope that occurred during storm Hilde on 15–16 November 2013. More than 100 landslides were registered during the storm. Rainfall totalled about 80–100 mm in 24 hours, locally up to 129 mm, and an additional trigger factor for the slides was a rapid rise in air temperature that caused snowmelt. On 15 November, the groundwater level in the hillslope rose by 10 cm per hour and reached 44 cm below the surface. At the same time, air temperature rose from 0 °C to over 8 °C, and the groundwater temperature dropped by 1.5 °C. The debris flow probably occurred late in the evening of 15 November, when the groundwater level reached its peak. Measurements of the groundwater in the hillslope in the period 2010–2013 show that the event in 2013 was not exceptional. Storm Dagmar on 25–26 December 2011 caused a similar rise in groundwater level, but did not trigger any failures. The data suggest that during heavy rainstorms the slope is in a critical state for a slide to be triggered for a short time – about 4–5 hours.

## 1 Introduction

It is well known that groundwater pore pressure is crucial in triggering shallow debris slides and flows (e.g. Iverson, 1997), but how exactly does pore pressure vary in a hillslope during a rainstorm? How much and how rapidly does the groundwater level rise before a slide is triggered? And, for how long during a rainstorm is the slope landslide-prone? We try to answer these questions using groundwater level data logged by an automated piezometer installed in a borehole on a hillslope in western Norway, where a debris flow occurred during heavy rainfall and snowmelt in November 2013.





Such instrumental data are rare because it is difficult to predict which slope will fail. Several studies have measured the hydrologic response to rainstorms in hillsides prone to shallow slides (e.g. Collins et al., 2012; Fannin and Jaakkola, 1999; Johnson and Sitar, 1990; Sidle, 1986), or measured the response to artificial sprinkling of water over a slope to force a slide to occur (e.g. Harp et al., 1990; Reid et al., 1997). Only a few studies provide instrumental data directly from a natural debris flow event (Montgomery et al., 2009; Reid et al., 1988). Here, we present continuous groundwater level measurements from

a hillside susceptible to shallow debris slides and flows for the period 2010–2013. The data include the day when a debris flow occurred in this particular slope during the storm named Hilde on 15–16 November 2013.

During this storm, the intense rainfall and snowmelt triggered more than 100 landslides in Western Norway. The maximum rainfall intensity was 80–100 mm in 24 hours, locally up to 129 mm. Most of the slides were debris slides and flows

(114), but rockfalls (28) and snow avalanches (7) also occurred (Fig. 1). Many roads were damaged, a bus got stuck, people were evacuated, and houses and cars were damaged by slide material, but fortunately no one was killed (Fig. 2). The number of slides makes this one of the largest landslide events in Norway during the last two decades (Krøgli et al., 2018).

The storm was a classic extreme precipitation event on the west coast of Norway. A warm mature front, partly occluded, passed southern Norway on 15 November. The pressure configuration, with a very low-pressure system northeast

of Jan Mayen and a high-pressure center southwest of the UK, generated a strong north-south pressure gradient and induced a southwesterly flow of moist warm air towards western Norway (Fig. 3). It rained heavily as the moist air was lifted by the warm front. The precipitation further intensified as the flow ascended over the mountains of western Norway. This weather situation leads to the formation of narrow plumes of intense high-level moisture, often referred to as atmospheric rivers. These cause most of the extreme precipitation events on the west coast of Norway (Azad and Sorteberg, 2017).



**Figure 1: Landslides that occurred on 15 and 16 November 2013 during storm Hilde in Western Norway – 114 debris flows, slides, 7 snow avalanches and 28 rockfalls. Data retrieved from the Norwegian landslide database: https://gis3.nve.no/link/?link=SkredHendelser. Precipitation is interpolated from observations and shows the amount in mm per 24 hours, collected at 06:00 UTC 16 November (08:00 local time), data retrieved from http://www.xgeo.no/.**









**Figure 2: Photos of slides that occurred during storm Hilde in 2013.**
**A: Skredestranda. The slide occurred at around 21-22 on 15 November, was triggered at an altitude of 620 m, covered about 250 m of the road and continued into lake Hornindalsvatnet (altitude 53 m). Photo: Jan Helge Aalbu, 16 November 2013.**
**B: Anestølen, this study. Photo: Kevin Saurin, 6 October 2014.**
**C: Oldedalen at Yri. The slide occurred around 23 on 15 November, started at about 350 m above sea level, and continued into lake Oldevatnet (altitude 34 m). Photo, Jan Helge Aalbu on 16 November 2013. The slides in A and C both generated small tsunamis.**

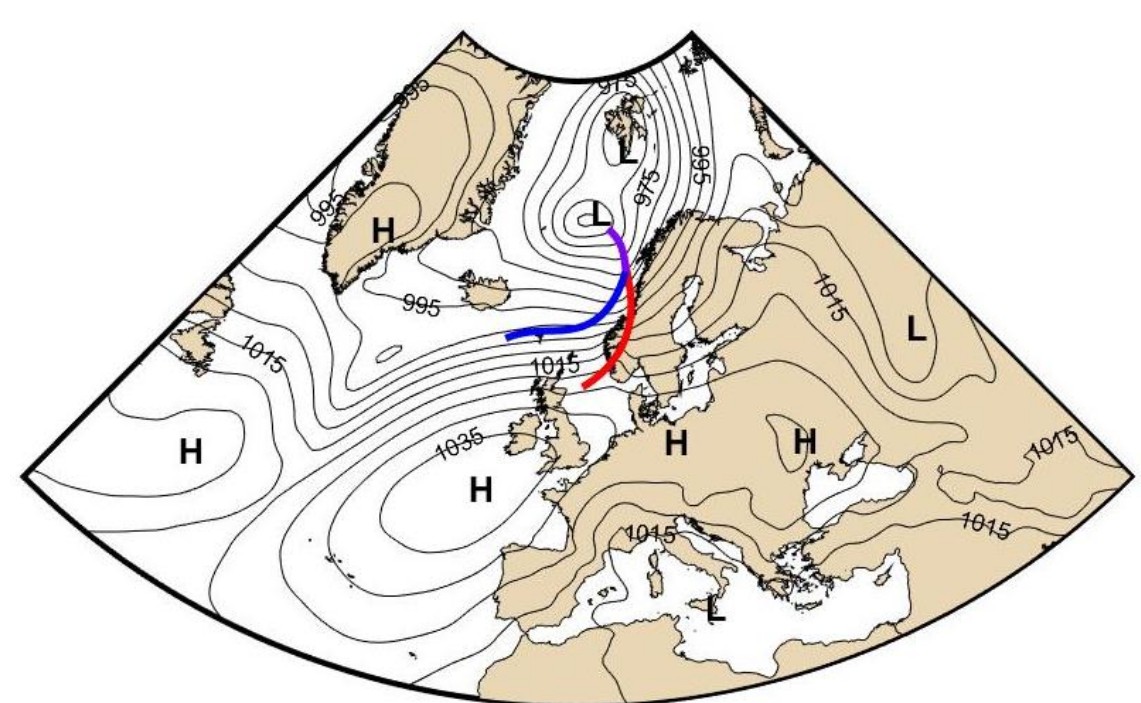

**Figure 3: Mean sea level pressure (hPa) on 15 November at 12:00 UTC (14:00 local time). Large amounts of warm and moist air blew towards the Norwegian coast because of low pressure northeast of Jan Mayen and high pressure southwest of England. The air was close to 100 % saturated with water vapor, which precipitated as the air cooled when it rose over the mountains in western Norway. The red line indicates the position of the warm front and the blue line is the cold front. The warm air sector lies between the red and blue lines. The purple line indicates an occluded front, which means that the cold front has caught up with the warm front.**



## 2 The hillside and instrumentation

The hillside we instrumented is typical of Western Norway, and is prone to debris flows and shallow slides. The hillside is near Sogndal, east of the lake at Anestølen (Fig. 4). Recent slide events on this slope occurred on 18 May 2004, 22 September 2007 (Tyssebotn and Velle, 2010) and 15 November 2013 (Olsen et al., 2015) (Fig. 4B). The lower part of the slope is covered by slide deposits and till and the average slope angle is 25 °– 26 °. The sediment-covered slope tapers off upwards into steeper and exposed bedrock and cliffs (Fig. 5A). From the outcrops of the slide scar of the 2007 slide event (Fig. 4B), we found that the thickness of the deposits on the slope varies, probably between 2 and 5 m.

We drilled through boulders and relatively firm deposits down to 2.4 m below the surface using a hammer drill powered by compressed air (Fig. 5B). We pushed and hammered threaded pipes, 1 m long and 32 mm in diameter, down into the borehole. The pipes were subsequently screwed together. The lower pipe (piezometer) has twelve slits, 10 cm long and 2 mm in width, to allow water seepage. The data logger, a mini-diver (DI 501) manufactured by Van Essen instruments, was attached to a wire and inserted into the pipe. The lower end of the mini-diver is at 1.64 m below the ground surface (Fig. 5B). The drilling and instrumentation are described in Tyssebotn and Velle (2010).

The weather station at Anestølen is operated by the Hydrological Department at the Norwegian Water Resources and Energy Directorate and measures air temperature, precipitation, wind, atmospheric pressure, snow depth and groundwater level (Fig. 4B). Precipitation is measured every 10 minutes, but for periods the bucket that collects the rain was full, and the data are not reliable. However, we found the measurements of precipitation to be reliable around the time of storm Hilde (Olsen et al., 2015). The other precipitation data we present are 24-hour totals measured manually at Selseng (Fig. 4A), 3.5 km south of Anestølen at about the same altitude (station 55730, data downloaded from http://eklima.met.no). Atmospheric pressure and air temperature (Fig. S1) at the weather station are measured every 15 minutes. All data from the weather station are available from http://sildre.nve.no (station no. 77.24.0).

The mini-diver we installed in the hillslope piezometer measure total pressure and temperature. Pressure is measured in cm $H_2O$ to an accuracy of ± 0.5 cm $H_2O$ with a resolution of 0.2 cm $H_2O$. Temperature is measured in degrees Celsius to an accuracy of ± 0.1 °C, with a resolution of 0.01 °C. The atmospheric pressure was subtracted from the measured pressure to obtain the true water pressure above the sensor in the mini-diver. We used the atmospheric pressure measured at the weather station from 2 October 2012 onwards (data downloaded from http://sildre.nve.no/Sildre/Station/77.24.0 ). Measurements from before 2 October 2012 were corrected using atmospheric pressure measured at the airport in Sogndal (station 55700, downloaded from http://eklima.met.no). The difference in altitude between the weather station (442 meters above sea level) and the sensor in the borehole at the slope (484 m) is 42 m. This corresponds to a pressure difference of about 5 hPa, which we subtracted from the atmospheric measurements. The difference in altitude between the airport (497 m) and the borehole is





only 13 m. For these measurements we ignored the altitude difference. The mini-diver was set to record measurements every
4 hours, and we obtained almost continuous measurements for the years 2010, 2011, 2012 and 2013.

**3 The 2013 debris flow**

The slide happened in the evening of 15 November or during the following night. It was first observed by a person
driving an all-terrain vehicle to the mountain farm at Anestølen in the morning of 16 November; he found that the road was
covered by wet debris, and was unable to proceed (Fig. S2). The driver thinks the flow happened sometime during the night,
but cannot exclude the possibility that it was the evening before, on 15 November (Olsen et al., 2015). The nearest slides to
Anestølen (see three triangles in Fig. 1 immediately west of this site), happened on the evening of 15 November, and during
the night and early morning of 16 November at 02:00 and 07:30.

The slide is a typical debris flow with levees and lobes. The flow had a point source at an altitude of 720 m, widened
downslope, and continued in a channel with levees on both sides (Figs. 2B, S3). The channel was eroded about 2 to 3 m into
the deposits, and exposed the bedrock in certain places. A spillover lobe is evident on the southern side of the slide. Towards
the lake, the flow split into three different paths (Fig. 2B) (Olsen et al., 2015), and ended in the lake (441 m altitude). Here, a
fan and a delta of fine-grained material, mostly sand and organic material, were deposited (Fig. S4). We inspected the slide on
17 November (Fig. S5) and found snow inside the very wet slide debris. The entire lake was colored brown by suspended
sediment from the slide that had entered the lake.


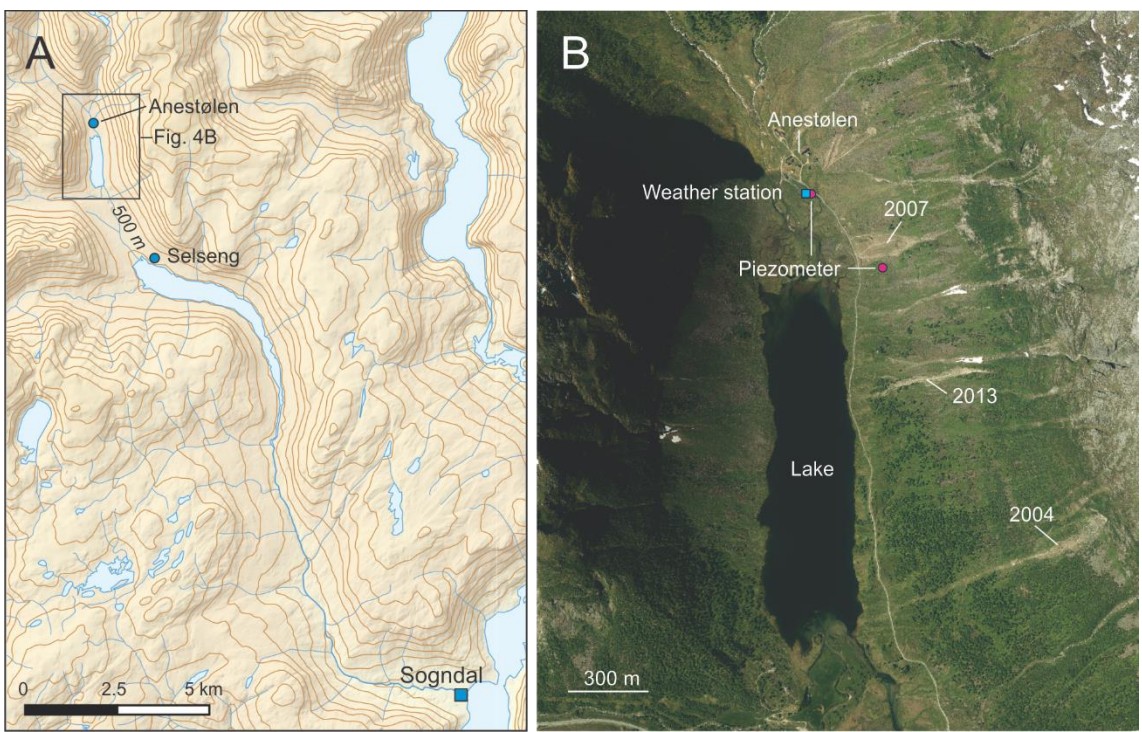

**Figure 4: Map and aerial photo of the site near Anestølen. A. The contour interval is 100 m. B. Aerial photo from 2018. The eastern slope is prone to debris slides and flows, and the most recent slides are indicated with the year they occurred. There is a weather station including a piezometer located in the valley bottom. We installed a piezometer in a borehole on the hillslope close to the southern slide scar of the 2007 slide. © Kartverket/Geovekst (www.kartverket.no).**

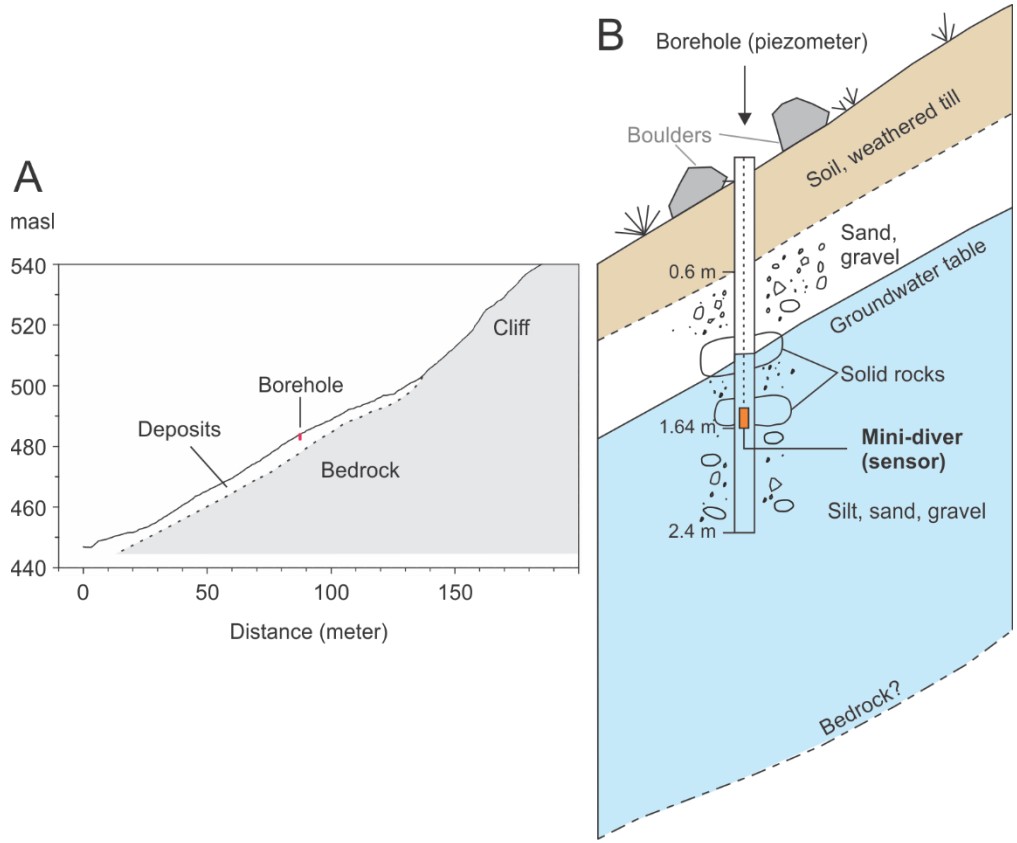

**Figure 5: Profile of the hillside showing the borehole with the instrumented piezometer. A. Average steepness of the sediment-covered slope is 25–26°. The stippled line indicates where we believe the boundary between till and bedrock is located, based on observations from the 2007 slide scar (Fig. 4B). B. The mini-diver (sensor) is suspended from a wire in the piezometer and anchored 1.64 m below the surface. The top 60 cm of deposits consists of organic soil and weathered till, with firm till below that. We drilled through boulders at depths of ca. 1 m and 1.5 m.**

## 4 Results

The groundwater level on the slope varies according to season and precipitation. In winter the groundwater table is low, and in 2013 the water level was below the sensor in February and March, apparently because the ground was frozen (e.g. Ireson et al., 2013) (Fig. 6). During spring, from April to June, the groundwater is recharged by snowmelt, rain and thawing of the frozen ground (Du et al., 2019), which cause high peaks and rapid changes. In summer and fall, the groundwater level is controlled by precipitation events, and normally oscillates between 150 cm and 75 cm below the ground (Figs. 6, S6–S8). A few times in the summer, the groundwater level is lower than the sensor. Most of the oscillations show a rapid, almost instantaneous rise, and a longer, slower, decline – the oscillations are clearly asymmetric (Fig. 7).





The peaks and troughs of the oscillations occur simultaneously upslope and in the valley bottom (piezometer at the
weather station), but the peaks upslope are much sharper and vary in amplitude (Fig. 7). The valley bottom is saturated almost
every time there is a precipitation event. The upslope peaks reach different heights, depending on the amount of infiltration
(Fig. 7). Another difference is in the shape of the peaks. The upslope maxima are very sharp, lasting less than 4 hours, while
the maxima downslope are broad. The peaks dissipate more rapidly upslope than downslope, as has also been described
elsewhere e.g. Alaska (Sidle, 1984, 1986). The groundwater level starts to decline within hours after reaching peak levels,
while the valley bottom peaks last for half a day or more (Fig. 8).

Groundwater temperatures follow the season, but show small, irregular oscillations that reflect infiltration episodes
(Figs. 6, S6-S8). The big picture is a steady, slow temperature decline through the winter to about 1.5 ° – 2 °C, an abrupt drop
in the spring when temperature reaches its minimum (0.5 ° – 1.2 °C) with wiggles in the temperature curve reflecting meltwater
reaching the sensor, followed by a large rise from May to August, up to 9.5 °– 10 °C. In September the groundwater
temperatures start a steady decline towards the winter minimum. However, the temperature curves show a number of small
anomalies that coincide with peaks in groundwater level. These anomalies are caused by infiltration of surface water that is
either colder or warmer than the groundwater. Infiltration episodes between October and May cause the temperature to drop;
the largest observed were drops of 1.5 °C and 1.8 °C  related to snowmelt (Figs. 6, S7). Between May and October, surface
water is warmer than the groundwater; infiltration episodes cause temperature rises, the largest recorded being about 1 °C and
associated with summer rain events in July (Fig. S7) and August (Fig. 6).

One of the largest oscillations in both groundwater level and temperature occurred during storm Hilde on 15–16
November 2013, and triggered the slide event. In only 8 hours, the groundwater level rose by 46 cm and the temperature
dropped by 1.5 °C (Fig. 8). The precipitation at Anestølen began at 08:20 on 15 November and fell as snow until about 14:30,
in all 22 mm (3.3 mm/h), because the air temperature was between 0 °and 0.6 °C (blue line in Fig. 8). From about 15:00 the
air temperature rose quickly and reached 9.5 °C at midnight (Fig. 8), and precipitation was 4.2 mm/h until 22:30 p.m. The
response in groundwater level was a rapid rise, 9.5 cm/h (Fig. 9), and a drop in temperature, from 4.9 °C to 3.4 °C between
15:00 and 19:00.

Prior to the storm event, there was a thin cover of snow on the ground. The snow pillow at the weather station
measured 20 mm of snow water equivalent before the storm began (Fig. S9). Photos of the slide the morning after the storm
show patches of snow on the ground (Figs. S2-S4). We also observed lumps of snow inside the slide debris. This suggests that
there was about 10-20 cm of snow on the ground prior to the arrival of warm, moist air with the storm.





**Figure 6: The upper diagram shows groundwater level (blue curve) and groundwater temperature (black curve) from the borehole on the slope in 2013. The lower diagram shows 24h precipitation at the weather station at Selseng (Fig. 4A), station no. 55700, http://eklima.met.no. One of the most pronounced oscillations is on 15–16 November, when a**
**debris flow occurred on this hillslope.**



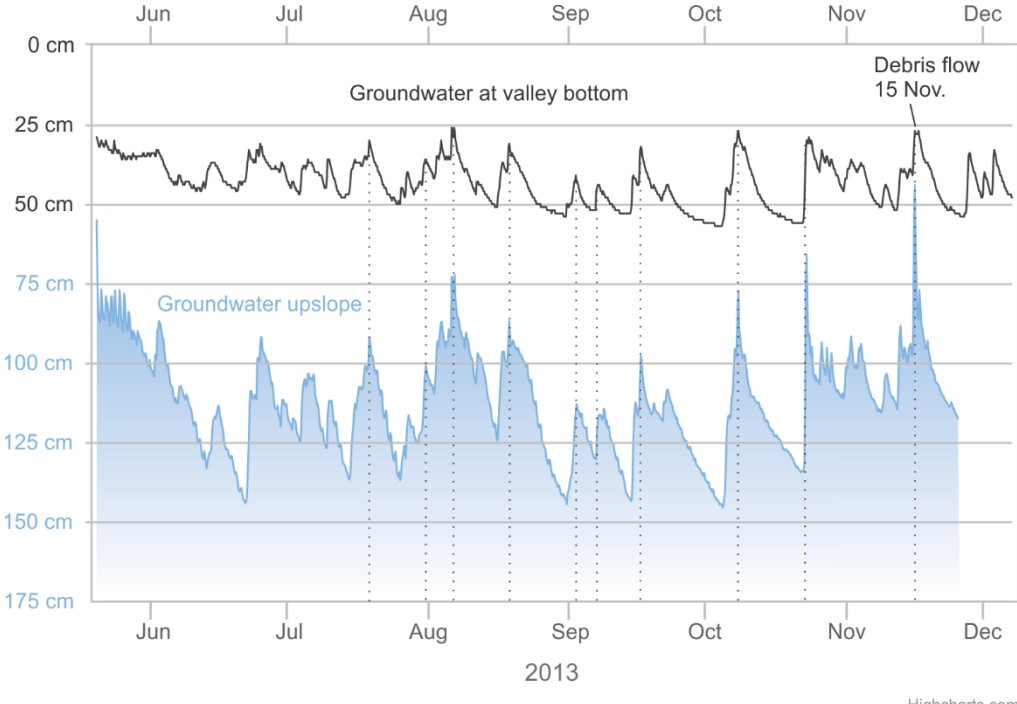

**Figure 7: Groundwater fluctuations in June-December 2013. The upper curve (black) shows fluctuations at the weather station in the valley bottom. The blue curve shows the fluctuations in the borehole upslope. The minima and maxima occur at the same time (dotted lines), but the peaks upslope are sharper, last less than 4 hours, have higher amplitudes, and the individual peaks reach different levels. The ground in the valley bottom is saturated at a registered depth of around 25 cm.**



**Figure 8: Air temperatures, precipitation and groundwater between 14 and 18 November 2013. The upper diagram shows air temperature (red curve) and precipitation in mm per 10 min. The lower diagram shows the groundwater level and groundwater temperature. Blue segments of the air temperature-curve show periods when the temperature is close to 0°C. Groundwater probably reached its highest level between 19:00 and 23:00.**




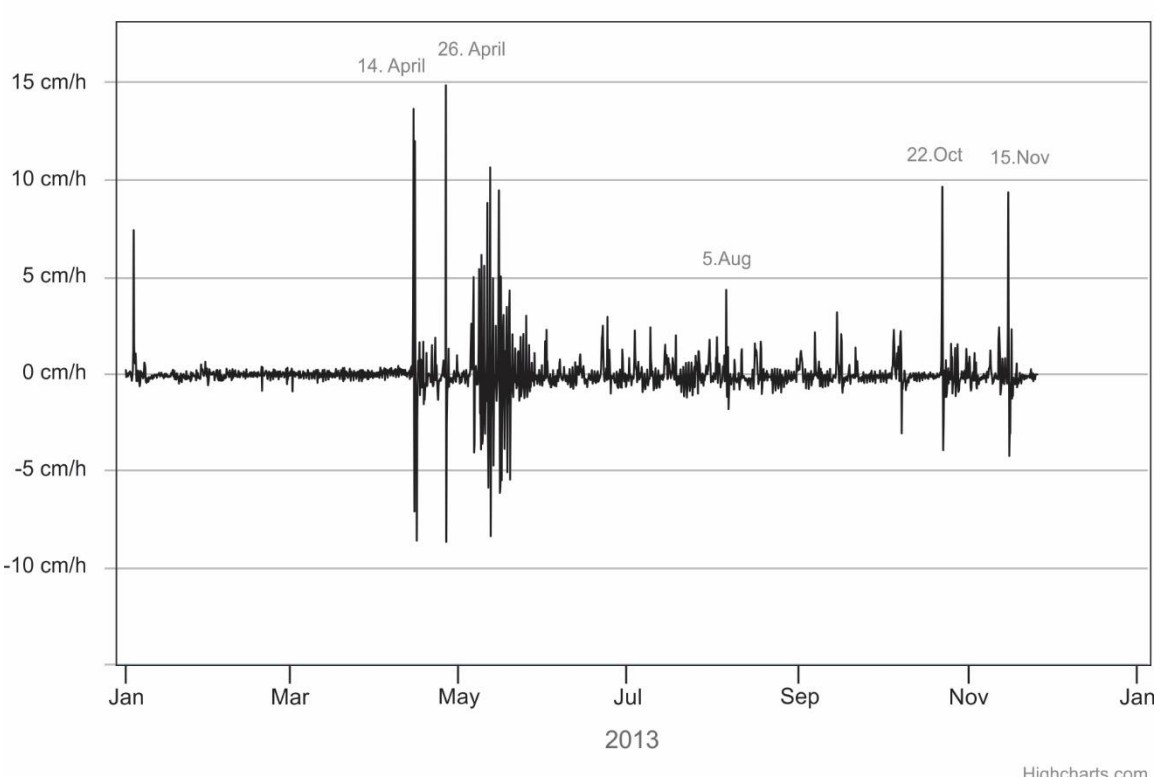

**Figure 9: Rates of change of groundwater level in 2013. The highest rates are in April and May and are caused by melting of the snow cover and thawing of the ground.**




## 5 Discussion

One weakness of this study is that the measurements on the slope are from one piezometer only. The hydrological response to the rainfall depends on antecedent moisture conditions and the porosity, hydraulic conductivity and thickness of deposits on the slope (Johnson and Sitar, 1990; Montgomery et al., 2009). The slope is covered by till and colluvium, deposits

that vary widely in composition and grain size. Thus, observations from one borehole may not be representative of other areas on the same slope (Fannin and Jaakkola, 1999). The distance from the borehole to the 2013 slide is about 400 m (Fig. 4B). Another weakness is that the piezometer in the hillslope was set to a recording interval of 4 hours, which is too long to capture details of the most rapid changes (Fannin and Jaakkola, 1999). The strength of the study is the continuous measurements of both pore pressure and groundwater temperature in conjunction with the nearby weather station over a period of 4 years that

also covers a slide event.

### 5.1 The storm Hilde landslide event, 15–16 November 2013

The groundwater level in the piezometer on the slope peaked between 19:00 and 23:00, and this is probably the time window when the debris flow was triggered. We measured the peak at 23:00 at 44 cm below surface, but because the

piezometer only took measurements every 4 hours, and the rise of the groundwater declined between 19:00 and 23:00 without any change in precipitation or air temperature, it is likely that the actual peak was reached earlier than 23:00 and was higher than 44 cm (see Fig. 6). Extrapolations of the groundwater level curve from 19:00 and 23:00 indicates that the peak might well have been at a depth of 30 cm and sometime around 20:30-21.30 (Fig. 8). From a study in Oregon, USA, Montgomery et al. (2009) concluded that most of their piezometers recorded that groundwater was still rising at the time of a debris flow failure,

and this suggests that the debris flow at Anestølen occurred earlier than 23:00.

The substantial drop in groundwater temperature suggests that part of the infiltration was from snowmelt. There was already some snow on the ground on 15 November (Fig. S9), and the precipitation in the morning and until about 15:00 fell as snow because of the low air temperature (Fig. 8). During this time there was no response in the groundwater level or

temperature. However, after 15:00, when the warm, moist air of the storm reached Anestølen, the groundwater rose by 9.5 cm/h and at the same time, the temperature of the groundwater dropped by 0.2 °C/h. The rapid snowmelt added extra water to the slope and augmented the infiltration rate and groundwater rise, as seen from rainfall on snow on slopes in Alaska (Sidle, 1984) .

The groundwater level in the hillslope continued to rise as long as the infiltration rate was higher than the downslope discharge of groundwater (Fig. 10, middle panel). The heavy rain ceased at 22:30, and after 23:00 the groundwater level fell by 3 cm/h (Fig. 10, lower panel). The groundwater level was still rising just after 19:00, so the peak itself must have been





shorter than 4 hours. Such a short peak suggests that very little groundwater from higher up the slope or from bedrock fractures (Johnson and Sitar, 1990) could have contributed to the rise in groundwater level. The entire groundwater rise can be explained

by vertical infiltration through the surface. The full amplitude of the groundwater peak was 54 cm during storm Hilde (Fig. 8), and this is an order of magnitude larger than the total amount of rainfall and snow (53 mm). A similar relationship has been found in other studies (e.g. Sidle, 1984).

### 5.2 Other groundwater episodes

Melting of the snow cover on the slope leads to high groundwater levels. In 2013 we recorded the year's highest peaks in April and May (Fig. 6), when the ground was still covered by substantial amounts of snow (Fig. S9). The high peaks correspond to lows in the groundwater temperature curve, reflecting the infiltration of cold meltwater. The small wiggles in the temperature curve end on 26 May – the same day the snow pillow at the weather station was free of snow (Fig. S9). The highest peaks were on 15 April and 16 May, 33 cm and 28 cm below the ground respectively, and coincided with rain (Fig. 6)

and high air temperatures (> 8°C) (Fig. S1). This high pore pressure did not cause any sliding, perhaps because there was still considerable snow cover, estimated at around 1 m (Fig. S9).

The fluctuations that occurred during storm Dagmar on 26–27 December 2011 were exceptional and similar to those during storm Hilde event. During storm Dagmar, groundwater rose to 28 cm below the surface, and the groundwater

temperature dropped by 1.8°C (Fig. S7). The groundwater rose by 13.5 cm/h between 19:00 and 23:00 on 26 December (Fig. S11). A major cause was a considerable increase in air temperature that melted a lot of snow, in combination with some precipitation (37.5 mm; Fig. S7). In spite of this, there were no observations of any sliding on this particular slope.

Another episode when there was a very rapid rise in groundwater level occurred on 18–19 August in 2012. Here, the

groundwater rose from below the sensor and up to 97 cm below the surface (Fig. S6) at a rate of more than 15.7 cm per hour (Fig. S10). This was the highest rate measured during the four years of recordings (Figs. 10, S10–S12). Nevertheless, no slide was observed, probably because the groundwater level was too low when the rain started, and so the peak did not reach higher than 97 cm below the surface. This shows that it is not the rise itself, but the level that the groundwater reaches, that is important.


When the groundwater level in the piezometer rises above 50 cm below the surface, the slope reaches a critical condition. Such a high groundwater level was only recorded three times during the four years of observations (2010-2013): during snow melting in April and May 2013 and during the storms Hilde in 2013 and Dagmar in 2011. On these occasions parts of the slope might have been fully saturated, or artesian conditions might exist, especially in depressions or areas where

conductive layers are thinning out or are blocked by less permeable deposits (Johnson and Sitar, 1990; Sidle, 1984). In order





for the pore pressure to reach such a high level during a rainstorm, the groundwater level before the onset of the storm has to be sufficiently high, not much deeper than 1 m below the ground. In addition, our data indicate that the groundwater level will only be above 50 cm for a very short time, probably less than 5–6 hours, and that these conditions only persist as long as infiltration is higher than the downslope discharge of groundwater. This pattern is in contrast to groundwater measurements from the nearby weather station in the valley bottom, which show much broader peaks.

### 5.3 Future changes in precipitation and snowmelt?

Many of the landslides triggered during storm Hilde were caused by simultaneous heavy rainfall and strong snowmelt. Such rain-on-snow events in Western Norway may become more frequent in the future. Storm Hilde was a typical atmospheric river storm, a type that has caused 57 of 60 extreme daily precipitation events in Western Norway since 1900. Of these, 62 % occurred in the months November, December and January – while none occurred in April, May, June and July (Azad and Sorteberg, 2017). The frequency of extreme precipitation events over Norway has increased by 25–35 % over the last 120 years (Sorteberg et al., 2018) and modelling indicates that it will increase further in the future (Hanssen-Bauer et al., 2017). Since most of these events occur in the winter months, it is very likely that we will see more rain-on-snow events in Western Norway in the near future that could increase the risk of slide events and floods.



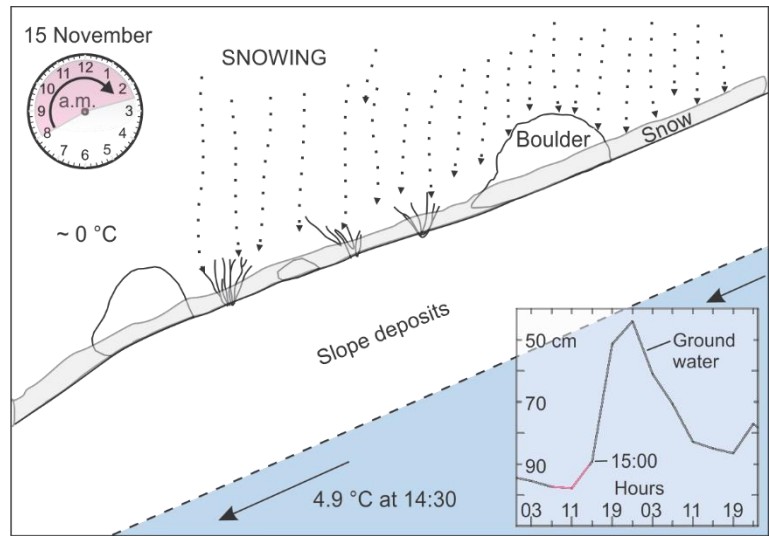

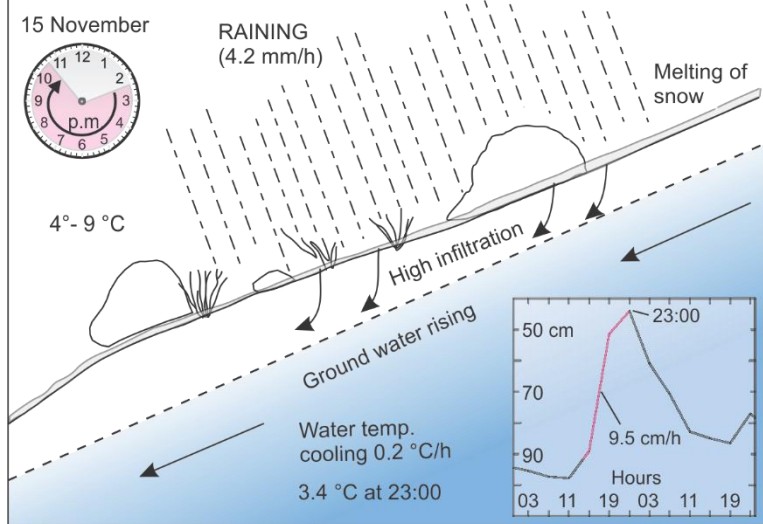

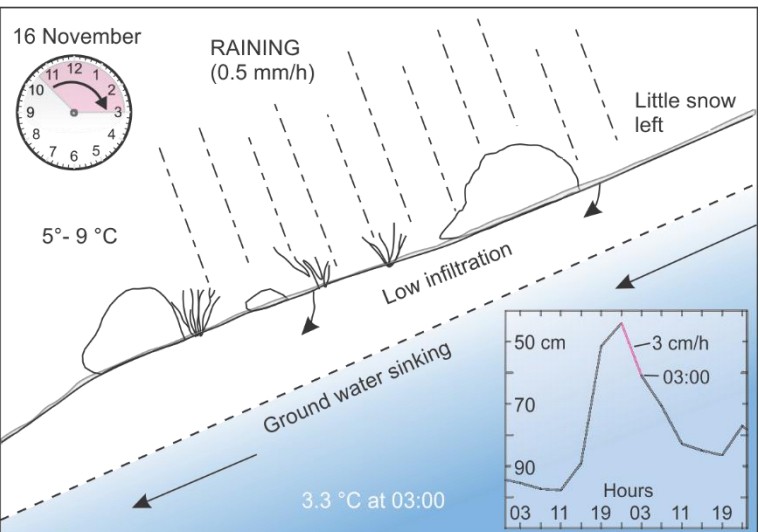





**Figure 10: Diagrams to explain the situation on the hillslope during storm Hilde on 15–16 November 2013. The purple**
**section of the clock and purple segment of the groundwater curve in the graph in the lower right corner show the time.**
**Upper panel: Between 08:00 and 14:30, air temperatures were close to 0° C and the precipitation fell as snow.**
**Groundwater level was stable (graph in lower right corner). Middle panel: Between 14:30 and 22:30, the air**
**temperature rose to 9 ° C, it rained 4.2 mm/h, groundwater level rose by 9.5 cm/h and groundwater temperatures**
**dropped by 1.6 ° C. Lower panel: After 22:30, the heavy rain ceased and the groundwater dropped by 3 cm/h.**





# 6 Conclusions

1.  The storm Hilde event in western Norway produced relatively large amounts of precipitation on the western slopes of the mountain range near the coast. Farther inland, precipitation was high but not extreme. The strong warm front in the storm gave rise to a rapid temperature increase of 8–9 °C, initiating snowmelt that supplemented the rainfall and lead to a rapid rise in groundwater. This situation triggered over hundred slides.

2.  We measured groundwater levels in a hillslope that failed in the storm. The groundwater responded rapidly to the rainfall and increase in air temperature during the storm and rose by 9.5 cm/h, simultaneously with a pronounced drop in groundwater temperature of 0.2°C/h. The groundwater peak reached at least 44 cm below the surface and the temperature dropped by 1.6 °C.

3.  The slope remained saturated or near saturated for a short time. Critical conditions of the slope lasted only for 4–5 hours during the Hilde storm.

4.  Two episodes stand out from the data collected over four years (2010, 2011, 2012 and 2013) in the hillslope piezometer; – the groundwater peaks during storm Hilde in 2013 and storm Dagmar in 2011, both storms had ground water levels that reached higher than 50 cm below the surface. Normally the groundwater level fluctuated between 150 cm and 75 cm below the surface. The groundwater level in the hillslope piezometer was below 50 cm for 99.87 % of the four years of data.

5.  The infiltration of surface water is clearly recorded in the groundwater temperature curve as a short-lived anomaly. The anomaly depends on the amount of infiltration and the temperature difference between the ground and the air. The largest negative anomalies are related to snowmelt in the fall, 1.5–1.8 °C. Summer rain caused at most a 1 °C positive temperature rise.

**Author contribution**

SB conceived the project, did the fieldwork, collected the data and wrote the first draft. AS described the weather situation during the storm (Fig. 3), was responsible for the precipitation map in Fig. 2 and wrote the first draft of chapter 5.3 "Future changes in precipitation and snowmelt."



**Declaration of competing interest**

The authors declare that they have no conflict of interest.

**Acknowledgments**

Ola Olsen, Kevin Saurin and Sondre Wenaas mapped the 2013 debris flow event at Anestølen and collected information about the slide event. Geir Magne Tyssebotn and Julia Heggdal Velle drilled the borehole on the hillslope and installed the piezometer with the first mini-diver. Knut Møen (Norwegian Water Resources and Energy Directorate) answered questions about data from the weather station. Ottar Husum provided local information. Jan Helge Aalbu allowed us to use his photos in Fig. 2A

and 2C. We inspected and plotted the data using the program Highcharts (https://www.highcharts.com/). Denise Christina Rüther and Helge Henriksen provided critical comments that improved the paper, and Alison Coulthard provided assistance with language editing.







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
