# Peer review of "Groundwater fluctuations during a debris flow event in Western Norway – triggered by rain and snowmelt"

_Hydrology and Earth System Sciences, 2020_

## Referee Comment (RC1) · Ivan Vergara (Referee) · 15 Sep 2020

GENERAL COMMENTS:

The manuscript "Groundwater fluctuations during a debris flow event in Western Norway – triggered by rain and snowmelt" analyses the behaviour of the groundwater level during the occurrence of a debris flow and compares it with the behaviour during other extreme events and with the typical groundwater situation. It is considered that the research is novel because the analysis of these data is not common and the results are very important for the study of shallow landslides by allowing to know the situation of the ground before, during and after the failure. Moreover, the manuscript is well

written and has fine and clear figures. Then are some comments that could improve understanding of the work.

SPECIFIC COMMENTS:

L19-20. Saying that Storm Dagmar did not generate debris flows without giving the reason does not provide much information to the reader. I think the sentence is missing a conclusion.

L82-87. It would be useful to describe somewhere in the manuscript (perhaps here) some other important characteristic of the study area... E.g., Mean annual precipitation, climate, vegetation.

L234-237. I don't understand the relationship between this sentence and Fig. 6. On the other hand, I do not understand the arguments to infer that the peak occurred before 23 o'clock... If the intensity of the rain and the air temperature did not change, it would not be explained either because the groundwater reached the peak between 8:30 p.m. and 9:30 p.m. and then it started to decrease.

L237-238. In Fig. 8 this extrapolation is not plotted... It is important to plot it.

L269-272. Is there any hypothesis why this event did not generate landslides? There is some information throughout the manuscript that could explain the non-occurrence and it could be useful to comment on them in the same paragraph (e.g., the 2013 peak could have been greater, the distance between where the debris flow is triggered and where the groundwater is measured, artesian conditions).

TECHNICAL CORRECTIONS:

L14-16. I think it should talk about precipitation and not rain, considering that until 3:00 p.m. the precipitation fell in a solid state; and only if it is possible (considering that in the Abstract each word counts) to clarify that a fraction of the melted snow came from the same event of precipitation.

L38-39. "The mean(?) maximum rainfall intensity was 80–100 mm in 24 hours, locally up to 129 mm." Currently the sentence is contradictory to me.

L39. "Most of the landslides were debris slides and flows" is clearer. Slide is a type of movement of landslides (Hungr et al. 2014) and using it as a synonym for Landslide can be confusing... The comment applies to the whole text.

L41-42. In order not to use the word "slide" and not repeat "landslide" it could be said something like this: "The number of mass movements makes this one of the largest landslide events in Norway during..."

L44-47. Many people are unfamiliar with Jan Mayen Island. With Fig. 3 it is clearer what its position should be, but it could be clarified in the text at least that it is an island.

L92. Is it necessary to clarify the brand?

L160. I understand that it is explained below but I don't think it is convenient to describe the shape ("much sharper") and not describe the difference in amplitude. I think it is better to either describe the two characteristics at the moment or just mention that they have different characteristics and then detail them in the following sentences.

L168. Perhaps a more technical term than "the big picture" can be used... Such as "the typical/mean annual cycle".

L252-253. Do you define peak duration as the time the groundwater was less than 50 cm from the ground? I think it is important to clarify.

L261-263. It may be helpful for the reader to indicate the date May 26 in Fig. 6.

Figure 1. Snow avalanches are not landslides and it is better to use the full terms... Landslide instead of slide and rockfall instead of rock.

Figure 1. A minor issue in the legend: the precipitation is a continuous number. This should also be reflected in the legend. Better write: 0-30; >30-60; >60-90;... etc.

Caption Figure 1. "114 debris flows and slides".

Figure 2. To respect the structure: "Photo: Jan Helge Aalbu on 16 November 2013."

Figure 4. Try to improve the sentence so as not to repeat "aerial photo". One possibility is: "Map and aerial photo from 2018 of the site near Anestølen. A. The contour interval is 100 m. B. The eastern slope is prone to. . ."

Figure 5B. It is not necessary to clarify that the rocks are solid.

———————————————————

---

## Referee Comment (RC2) · Anonymous Referee #2 · 29 Jan 2021

**GENERAL COMMENTS**

The manuscript provides continuous data about precipitation, air and groundwater temperature, snow depth, pore-water pressure, monitored on a site in Western Norway that in November 2013 was involved in a weather-induced debris flow after a storm that caused about 142 landslides and 7 snow avalanches in the same region. The reported data allow, in particular, to compare the weather and piezometric conditions responsible of the debris flow with those occurred in the past not able to induce any failure.

The paper is well written and contains very accurate figures. The availability of infor-

mation exactly during the landslide event represents undoubtedly a valuable aspect not to be overlooked. However, as also correctly recognized in the text by the Authors, two evident limitations exist: i) the pore-water pressure data are measured only by one piezometer (located upslope); ii) information about the properties of the involved soils are absent. Of course, the first aspect, that prevents to model the piezometric regime along the slope, can not be solved. On the contrary, I hope that some data about the physical, hydraulic and mechanical soil properties should be added because a full comprehension of the landslide is very hard without them. In particular, the absence of information regarding the shear strength parameters makes impossible analyzing the slope stability conditions. Some specific suggestions, aimed to improve the quality of the manuscript, are reported in the following section.

SPECIFIC COMMENTS

Line 30. The availability of real-time water level data during rapid landslides are effectively rare, but, on the contrary, many papers provide the pore pressure in slopes involved in active slow landslides, cyclically reactivated by seasonal weather events. Therefore, the sentence "rare because it is difficult to predict which slope will fail" should be replaced by "rarely provided during rapid landslide events".

Lines 39-40. Snow avalanches are not landslides. Therefore, the sentence "Most of the slides were debris slides and flows (114), but rockfalls (28) and snow avalanches (7) also occurred" should be replaced by "Most of them were debris slides and flows (114), but rockfalls (28) also occurred. Some snow avalanches (7) were observed too".

Figure 1. The term "slides" in the legend can not be used to indicate at the same time the three types of phenomena. It should be replaced by a term like "Events". Moreover, I suggest to indicate them according to the following order: i) Debris flow and slide; ii) Rockfall; iii) Snow avalanche.

Caption Figure 2. I suggest to simplify it, inserting in a table (to be cited in the text) all the provided information regarding the three shown landslides: date and hour of

the occurring events, landslide length, upslope and downslope altitudes, mean slope inclination, range of thickness, etc.

Section 2. This section should contain a table reporting the available information (eventually deriving them by other papers) about the mean values of physical, hydraulic and mechanical properties of the involved soils: grain size, in-situ porosity and degree of saturation, unit weight, hydraulic conductivity, strength parameters. Such values are very important to allow a full understanding of the infiltration and seepage processes and, as a consequence, of the induced landslide mechanism. Line 90. Indicate at which altitude and distance from the toe of the landslide the piezometer has been installed.

Line 125. Indicate the total length of the debris flow.

Figure 5A. Clarify which "Distance" is reported in the X-axis. Is it the distance from the toe of the landslide ?

Figure 6. According to the results, the influence of the snow cover melting on the water level is particularly important. Therefore, I suggest to insert in this figure the data about the snow depth (shown only by the supplementary Figure S9) and about the air temperature (partially shown in Figure 8).

Caption Figure 7. Indicate at which depth and altitude the piezometer of the weather station has been installed.

Figure 8. Due to the important role of the snow melting, I suggest to insert in this figure the data about the snow depth (shown by the supplementary Figure S9).

Line 222. The second important weak point of the manuscript regards the absolute absence of information about the soil properties. As already suggested, I hope that you are able to provide them. For instance, some information about the strength parameters could help (at least) estimating the slope stability conditions.

Lines 265-266. Differently from what observed in November 2013, the piezometric

peaks monitored in April and May 2013 were caused only by rainwater infiltration (and not by snow melting). Why do you consider such evidence so relevant to not induce sliding? The corresponding measured peaks of 33 cm (measured in April) and 28 cm (measured in May) below the ground surface are very close to the critical estimated value of 30 cm in November 2013 (such value was extrapolated from the groundwater level curve measured from 19:00 and 23:00, as clarified in Lines 237-238). As a consequence, being the local shear strength approximatively the same at the onset of the three attained maximum water levels, the corresponding local slope stability conditions should be essentially the same too. Unfortunately, the availability of only one piezometer does not allow to make a reliable evaluation of the general slope stability conditions, therefore your consideration seems rather rash. Please make some comments.

Lines 276-279. The emphasis of provided considerations is rather strange. It's well known that the initial conditions are crucial to determine the weather-induced effects. Once given an initial monitored piezometric value, the main challenge should be, of course, associating a landslide hazard to a forecasted weather event. At the same time, associating a very low landslide hazard to severe weather event if the initial measured groundwater level is located below a "safe" value should be also very useful for the implementation of an early warning system. I encourage the Authors to make some comments about this topic.

TECHNICAL CORRECTIONS

Title. Is the hyphen "-", between "Norway" and "triggered", necessary?

Caption Figure 1. "114 debris flows, slides" should be replaced by "114 debris flows and slides"

Line 109. The word "from" at the end of the line should be replaced by "carried out".

Line 163. The sentence "has also been" should be replaced by "already".

264, 2020.

---

## Author Comment (AC1) · 26 Feb 2021

**Response from the authors to the comments by referee Ivan Vergara.**

GENERAL COMMENTS:

The manuscript "Groundwater fluctuations during a debris flow event in Western Norway – triggered by rain and snowmelt" analyses the behaviour of the groundwater level during the occurrence of a debris flow and compares it with the behaviour during other extreme events and with the typical groundwater situation. It is considered that the research is novel because the analysis of these data is not common and the results are very important for the study of shallow landslides by allowing to know the situation of the ground before, during and after the failure. Moreover, the manuscript is well written and has fine and clear figures. Then are some comments that could improve understanding of the work.

We appreciate that you find our paper well written with fine and clear figures, and also that you have provided many suggestions, listed below, that will improve our paper. We will comments on each of them below.

SPECIFIC COMMENTS:

L19-20. Saying that Storm Dagmar did not generate debris flows without giving the reason does not provide much information to the reader. I think the sentence is missing a conclusion.

The sentence gives the reader the important information that in spite of a very similar groundwater situation, a debris flow or slide was not triggered during the storm Dagmar. Unfortunately, we do not know the reason for that and cannot give it. In the discussion section of the paper we discuss this, but are not able to come up with a specific explanation.

L82-87. It would be useful to describe somewhere in the manuscript (perhaps here) some other important characteristic of the study area... E.g., Mean annual precipitation, climate, vegetation.

Yes, we will do so; add a few sentences that will describe the vegetation and climate in the area.

L234-237. I don't understand the relationship between this sentence and Fig. 6. On the other hand, I do not understand the arguments to infer that the peak occurred before 23 o'clock... If the intensity of the rain and the air temperature did not change, it would not be explained either because the groundwater reached the peak between 8:30 p.m. and 9:30 p.m. and then it started to decrease.

Sorry, there is a type error here in our text, it should not say "*see Fig. 6*" in our text, but it should be "*see Fig. 8.*"

L237-238. In Fig. 8 this extrapolation is not plotted... It is important to plot it.

Yes, we agree, good idea. We will plot the extrapolation in Fig. 8.

L269-272. Is there any hypothesis why this event did not generate landslides? There is some information throughout the manuscript that could explain the non-occurrence and it could be useful to comment on them in the same paragraph (e.g., the 2013 peak could have been greater, the distance between where the debris flow is triggered and where the groundwater is measured, artesian conditions).

Yes, we agree. We will give some ideas here to why there were no landslides triggered during the storm Dagmar.

TECHNICAL CORRECTIONS:

L14-16. I think it should talk about precipitation and not rain, considering that until 3:00 p.m. the precipitation fell in a solid state; and only if it is possible (considering that in the Abstract each word counts) to clarify that a fraction of the melted snow came from the same event of precipitation.

Yes, agree. We will change from *"rain"* to *"precipitation"* – and also clarify in the abstract that some of the melted snow came from the same event.

L38-39. "The mean(?) maximum rainfall intensity was 80–100 mm in 24 hours, locally up to 129 mm." Currently the sentence is contradictory to me.

Yes, we see that this phrasing is not good and we will rephrase it.

L39. "Most of the landslides were debris slides and flows" is clearer. Slide is a type of movement of landslides (Hungr et al. 2014) and using it as a synonym for Landslide can be confusing. . . The comment applies to the whole text.

Yes, we will replace the word "slides" to "landslides".

L41-42. In order not to use the word "slide" and not repeat "landslide" it could be said something like this: "The number of mass movements makes this one of the largest landslide events in Norway during. . ."

Yes, we agree, we will use the word "mass movements".

L44-47. Many people are unfamiliar with Jan Mayen Island. With Fig. 3 it is clearer what its position should be, but it could be clarified in the text at least that it is an island.

Thanks for pointing this out. Instead of relating the location of the low atmospheric pressure to the island Jan Mayen, we will refer to Iceland, most people are familiar with the location of Iceland. We will change our text to (changes in red text and strikethrough):

*"The pressure configuration, with a very low-pressure system northeast of Iceland  and a high-pressure center southwest of the UK, generated a strong north-south pressure gradient and induced a southwesterly flow of moist warm air towards western Norway (Fig. 3)."*

L92. Is it necessary to clarify the brand?

Maybe not. It is important to give information about the equipment used here in the method section. We do not think it is a necessary to mention the brand of the data logger we used, but it does not hurt to mention it either. Our sentence is: *The data logger, a mini-diver (DI 501) manufactured by Van Essen instruments, was attached to a wire and inserted into the pipe.*

L160. I understand that it is explained below but I don't think it is convenient to describe the shape ("much sharper") and not describe the difference in amplitude. I think it is better to either describe the two characteristics at the moment or just mention that they have different characteristics and then detail them in the following sentences.

We agree. We will change to:

*"The peaks and troughs of the oscillations occur simultaneously upslope and in the valley bottom (piezometer at the weather station), but the peaks upslope are much sharper and have higher  amplitudes (Fig. 7)."*

L168. Perhaps a more technical term than "the big picture" can be used... Such as "the typical/mean annual cycle".

Yes, we agree.

L252-253. Do you define peak duration as the time the groundwater was less than 50 cm from the ground? I think it is important to clarify.

Yes, we agree – this is important to clarify, and we will do so in the revised manuscript.

L261-263. It may be helpful for the reader to indicate the date May 26 in Fig. 6.

Yes, good idea. We will indicate May 26 in Fig. 6.

Figure 1. Snow avalanches are not landslides and it is better to use the full terms. . . Landslide instead of slide and rockfall instead of rock.

Yes, we agree. We will change the legend in Fig. 1.

Figure 1. A minor issue in the legend: the precipitation is a continuous number. This

should also be reflected in the legend. Better write: 0-30; >30-60; >60-90;. . . etc.

Yes, we agree.

Caption Figure 1. "114 debris flows and slides".

Yes, we agree. We will change from "114 debris flows, slides,…." to "114 debris flows and slides" – as suggested.

Figure 2. To respect the structure: "Photo: Jan Helge Aalbu on 16 November 2013."

Yes, we will change so the structure is the same.

Figure 4. Try to improve the sentence so as not to repeat "aerial photo". One possibility is: "Map and aerial photo from 2018 of the site near Anestølen. A. The contour interval is 100 m. B. The eastern slope is prone to. . ."

Good idea. We will change it as suggested. Instead of "solid rocks" say "rocks".

Figure 5B. It is not necessary to clarify that the rocks are solid.

Agree – we will change this in Figure 5B.

---

## Author Comment (AC2) · 26 Feb 2021

**Response from the authors to the comments by anonymous referee.**

GENERAL COMMENTS

The manuscript provides continuous data about precipitation, air and groundwater temperature, snow depth, pore-water pressure, monitored on a site in Western Norway that in November 2013 was involved in a weather-induced debris flow after a storm that caused about 142 landslides and 7 snow avalanches in the same region. The reported data allow, in particular, to compare the weather and piezometric conditions responsible of the debris flow with those occurred in the past not able to induce any failure. The paper is well written and contains very accurate figures.

We are glad you think the paper is well written and has accurate figures. We appreciate all your suggestions that will improve our paper. Below we comment on each of these.

The availability of information exactly during the landslide event represents undoubtedly a valuable aspect not to be overlooked. However, as also correctly recognized in the text by the Authors, two evident limitations exist: i) the pore-water pressure data are measured only by one piezometer (located upslope); ii) information about the properties of the involved soils are absent. Of course, the first aspect, that prevents to model the piezometric regime along the slope, can not be solved. On the contrary, I hope that some data about the physical, hydraulic and mechanical soil properties should be added because a full comprehension of the landslide is very hard without them. In particular, the absence of information regarding the shear strength parameters makes impossible analyzing the slope stability conditions. Some specific suggestions, aimed to improve the quality of the manuscript, are reported in the following section.

About the properties of the involved soils:

Ideally, it would be very nice to present the properties of the soils on the slope in a table. However, we do not have such data, such data are not available from other studies of the area, and the material on the slope varies a lot, both laterally and with depth, that makes it difficult. In the paper we present the information we have about the properties of the slope deposits that allow the reader to have some understanding of the slope and the soil. Below is line 85-90 in the method chapter:

*"The lower part of the slope is covered by slide deposits and till and the average slope angle is 25 °– 26 °. The sediment-covered slope tapers off upwards into steeper and exposed bedrock and cliffs (Fig. 5A). From the outcrops of the slide scar of the 2007 slide event (Fig. 4B), we found that the thickness of the deposits on the slope varies, probably between 2 and 5 m.*

*We drilled through boulders and relatively firm deposits down to 2.4 m below the surface using a hammer drill powered by compressed air (Fig. 5B).*

In the caption to Fig. 5B (line 143-146) we say:

*"Average steepness of the sediment-covered slope is 25 °–26 °. The stippled line indicates where we believe the boundary between till and bedrock is located, based on observations from the 2007 slide scar (Fig. 4B). B. The mini-diver (sensor) is suspended from a wire in the piezometer*

*and anchored 1.64 m below the surface. The top 60 cm of deposits consists of organic soil and weathered till, with firm till below that. We drilled through boulders at depths of ca. 1 m and 1.5 m."*

And, in the discussion chapter we say (line 224-226):

*"The slope is covered by till and colluvium, deposits that vary widely in composition and grain size. Thus, observations from one borehole may not be representative of other areas on the same slope."*

In order to provide accurate data about the soil properties, especially data for cohesion and friction angle (the shear strength parameters), and data of the hydraulic conductivity etc., we would need to do a new study of the slope. Because of the heterogeneity of the soil, we would need information from several locations along the slope and also from different depths. On an ideal slope with homogeneous soils such data would be very useful, and possible also available. In addition, in this paper, we do not perform any calculations or simulations of stability or of the slide movement. We believe that the information given is sufficient for the reader to get the information of the soils on the slope to understand the work and data we present about the groundwater fluctuations during the debris flow event.

SPECIFIC COMMENTS

Line 30. The availability of real-time water level data during rapid landslides are effectively rare, but, on the contrary, many papers provide the pore pressure in slopes involved in active slow landslides, cyclically reactivated by seasonal weather events. Therefore, the sentence "rare because it is difficult to predict which slope will fail" should be replaced by "rarely provided during rapid landslide events".

We agree, thanks for this suggestion, we will replace as you indicate here.

Lines 39-40.

Snow avalanches are not landslides. Therefore, the sentence "Most of the slides were debris slides and flows (114), but rockfalls (28) and snow avalanches (7) also occurred" should be replaced by "Most of them were debris slides and flows (114), but rockfalls (28) also occurred. Some snow avalanches (7) were observed too".

We agree. We will change "slides" with "mass movements". The sentence will then be changed to:

*"Most of the  mass movements were debris slides and flows (114), but rockfalls (28) and snow avalanches (7) also occurred (Fig. 1).*

Figure 1. The term "slides" in the legend can not be used to indicate at the same time the three types of phenomena. It should be replaced by a term like "Events". Moreover, I suggest to indicate them according to the following order: i) Debris flow and slide; ii) Rockfall; iii) Snow avalanche.

Yes, we agree and will change it as you suggested here.

Caption Figure 2.

I suggest to simplify it, inserting in a table (to be cited in the text) all the provided information regarding the three shown landslides: date and hour of the occurring events, landslide length, upslope and downslope altitudes, mean slope inclination, range of thickness, etc.

It seems as a good idea. We will provide such a table and simplify the caption to Fig. 2.

Section 2.

This section should contain a table reporting the available information (eventually deriving them by other papers) about the mean values of physical, hydraulic and mechanical properties of the involved soils: grain size, in-situ porosity and degree of saturation, unit weight, hydraulic conductivity, strength parameters. Such values are very important to allow a full understanding of the infiltration and seepage processes and, as a consequence, of the induced landslide mechanism.

See comment above under the heading "About the properties of the involved soils."

Line 90.

Indicate at which altitude and distance from the toe of the landslide the piezometer has been installed.

OK!

Line 125.

Indicate the total length of the debris flow.

OK!

Figure 5A. Clarify which "Distance" is reported in the X-axis. Is it the distance from the toe of the landslide ?

No, it is not, it is the distance from the road on the flat valley bottom and upslope to the borehole, at 90° to the slope. We will indicate the location of the profile on the aerial photo in Fig. 4B, that will clarify the location of the profile.

Figure 6. According to the results, the influence of the snow cover melting on the water level is particularly important. Therefore, I suggest to insert in this figure the data about the snow depth (shown only by the supplementary Figure S9) and about the air temperature (partially shown in Figure 8).

In an earlier version of the paper we had included the graph of the snow cover (now in Fig. S9) in Fig. 6. We will go back to this earlier version now and move the snow cover figure from the supplements to Fig. 6. We will also consider to move the air temperature graph from the supplements to the main paper.

Caption Figure 7. Indicate at which depth and altitude the piezometer of the weather station has been installed.

Yes, we will provide that information.

Figure 8. Due to the important role of the snow melting, I suggest to insert in this figure the data about the snow depth (shown by the supplementary Figure S9).

Melting of snow has an important role, but we are uncertain that the measurements from the snow pillow at the day the slide happened, with low values, can be fully trusted. The snow pillow show 0.020 m of water equivalent before the event, and during the event rises to 0.024 m. This value is a combination of snow that was on the ground before the event, new snow during the first phase of the event and a change to rain. We want to look more carefully at the snow data and will strongly consider to present a graph of the data from the snow pillow in Figure 8.

Line 222.

The second important weak point of the manuscript regards the absolute absence of information about the soil properties. As already suggested, I hope that you are able to provide them. For instance, some information about the strength parameters could help (at least) estimating the slope stability conditions.

Again, see comment above under the heading "Information about the properties of the involved soils"

Lines 265-266.

Differently from what observed in November 2013, the piezometric peaks monitored in April and May 2013 were caused only by rainwater infiltration (and not by snow melting). Why do you consider such evidence so relevant to not induce sliding ? The corresponding measured peaks of 33 cm (measured in April) and 28 cm (measured in May) below the ground surface are very close to the critical estimated value of 30 cm in November 2013 (such value was extrapolated from the groundwater level curve measured from 19:00 and 23:00, as clarified in Lines 237-238). As a consequence, being the local shear strength approximatively the same at the onset of the three attained maximum water levels, the corresponding local slope stability conditions should be essentially the same too. Unfortunately, the availability of only one piezometer does not allow to make a reliable evaluation of the general slope stability conditions, therefore your consideration seems rather rash. Please make some comments.

We agree; the very high peaks in May and April is essential the same values as the critical peak in November 2013, but we now think, thanks to this review comment that encouraged more thinking about this, that the soil on the slope might have been partly frozen in April and May, especially the slope downslope of the borehole, that could have prevented effective drainage of groundwater downslope. The thawing of the frozen soil, melting of snow on the ground and the rain episodes caused the high piezometric peaks. In spite of the high pore pressures in the borehole, the slope was maybe not unstable, because part of the soil on the slope was maybe still frozen, and a frozen pore space would give higher shear strength. The air temperatures in April and May was low, most of the

nights were below freezing until May 5, that would prevent the thawing of the ground and refreezing of the surface water during nights.

We will rewrite this paragraph to enlighten this possibility of frozen soils in April and May.

Lines 276-279.

The emphasis of provided considerations is rather strange. It's well known that the initial conditions are crucial to determine the weather-induced effects. Once given an initial monitored piezometric value, the main challenge should be, of course, associating a landslide hazard to a forecasted weather event. At the same time, associating a very low landslide hazard to severe weather event if the initial measured groundwater level is located below a "safe" value should be also very useful for the implementation of an early warning system. I encourage the Authors to make some comments about this topic.

We will rephrase this paragraph; it also seems to us to be somewhat immature. This is a nice example of an event that would not bring the early warning system to an alarm despite the extreme infiltration rate. Thanks for pointing this out for us, we will try to rephrase this.

TECHNICAL CORRECTIONS

Title. Is the hyphen "-", between "Norway" and "triggered", necessary ?

It is not necessary for us to use "–", a comma instead would also work. However, using "–" (em dashes) to replace commas makes the reader focus a bit more on this information that is set inside the em dashes "–".

Caption Figure 1. "114 debris flows, slides" should be replaced by "114 debris flows and slides"

Yes, we agree.

Line 109.

The word "from" at the end of the line should be replaced by "carried out".

? We are not sure if we understand this comment …."carried out"…

Line 163.

The sentence "has also been" should be replaced by "already".

OK

---

## Author Response (AR1)

**Response from the authors (in blue) to the comments by referee Ivan Vergara (in black), updated 9. April 2021.**

[Line numbers in blue refers to the new version of the manuscript: Bondevik_Sorteberg_Groundwater fluctuations_revised_marked_up.docx]

GENERAL COMMENTS:

The manuscript "Groundwater fluctuations during a debris flow event in Western Norway – triggered by rain and snowmelt" analyses the behaviour of the groundwater level during the occurrence of a debris flow and compares it with the behaviour during other extreme events and with the typical groundwater situation. It is considered that the research is novel because the analysis of these data is not common and the results are very important for the study of shallow landslides by allowing to know the situation of the ground before, during and after the failure. Moreover, the manuscript is well written and has fine and clear figures. Then are some comments that could improve understanding of the work.

We appreciate that you find our paper well written with fine and clear figures, and also that you have provided many suggestions, listed below, that will improve our paper. We will comments on each of them below.

SPECIFIC COMMENTS:

L19-20. Saying that Storm Dagmar did not generate debris flows without giving the reason does not provide much information to the reader. I think the sentence is missing a conclusion.

The sentence gives the reader the important information that in spite of a very similar groundwater situation, a debris flow or slide was not triggered during the storm Dagmar. Unfortunately, we do not know the reason for that and cannot give it. In the discussion section of the paper this is somewhat discussed, but are not able to come up with a specific explanation.

L82-87. It would be useful to describe somewhere in the manuscript (perhaps here) some other important characteristic of the study area... E.g., Mean annual precipitation, climate, vegetation.

Yes, we have now included a paragraph that describes the vegetation and climate in the area (line 98-101).

L234-237. I don't understand the relationship between this sentence and Fig. 6. On the other hand, I do not understand the arguments to infer that the peak occurred before 23 o'clock... If the intensity of the rain and the air temperature did not change, it would not be explained either because the groundwater reached the peak between 8:30 p.m. and 9:30 p.m. and then it started to decrease.

Sorry, there is a type error here in our text, it should not say "*see Fig. 6*" in our text, but it should be "*see Fig. 8.*" This has now been changed (line 263).

L237-238. In Fig. 8 this extrapolation is not plotted... It is important to plot it.

Yes, we agree, good idea. We have now plotted the extrapolation in Fig. 8 as a stippled line.

L269-272. Is there any hypothesis why this event did not generate landslides? There is some information throughout the manuscript that could explain the non-occurrence and it could be useful to comment on them in the same paragraph (e.g., the 2013 peak could have been greater, the distance between where the debris flow is triggered and where the groundwater is measured, artesian conditions).

Yes, we agree, there must be a reason for this that has to do with local conditions in the slope. After reading the review the first time we agreed to give some possible explanations in our text, but we now feel it will just be speculations. We think it is more honest to say that we do not know. So, after thinking about this for some time we have decided to leave this paragraph as it is.

TECHNICAL CORRECTIONS:

L14-16. I think it should talk about precipitation and not rain, considering that until 3:00 p.m. the precipitation fell in a solid state; and only if it is possible (considering that in the Abstract each word counts) to clarify that a fraction of the melted snow came from the same event of precipitation.

Yes, agree. We have changed from "*rain*" to "*precipitation*" – and included a sentence that says that a fraction of the precipitation first came as snow (line 15).

L38-39. "The mean(?) maximum rainfall intensity was 80–100 mm in 24 hours, locally up to 129 mm." Currently the sentence is contradictory to me.

Yes, we see that this phrasing is not good, and we have rephrased it (line 40-42).

L39. "Most of the landslides were debris slides and flows" is clearer. Slide is a type of movement of landslides (Hungr et al. 2014) and using it as a synonym for Landslide can be confusing. . . The comment applies to the whole text.

Yes, we have replaced the word "slides" to "landslides" here (line 41) – but also at other places in our text where such a change was appropriate.

L41-42. In order not to use the word "slide" and not repeat "landslide" it could be said something like this: "The number of mass movements makes this one of the largest landslide events in Norway during. . ."

Yes, we agree, we have replaced slide with "mass movements" in line 45.

L44-47. Many people are unfamiliar with Jan Mayen Island. With Fig. 3 it is clearer what its position should be, but it could be clarified in the text at least that it is an island.

Thanks for pointing this out. Instead of relating the location of the low atmospheric pressure to the island Jan Mayen, we refer to Iceland, most people are familiar with the location of Iceland. We have replaced Jan Mayen with Iceland in our text (line 49) and also in the figure text (line 84).

L92. Is it necessary to clarify the brand?

Maybe not. It is important to give information about the equipment used here in the method section. We do not think it is a necessary to mention the brand of the data logger we used, but it does not hurt to mention it either. Our sentence is not changed: "*The data logger, a mini-diver (DI 501) manufactured by Van Essen instruments, was attached to a wire and inserted into the pipe.*"

L160. I understand that it is explained below but I don't think it is convenient to describe the shape ("much sharper") and not describe the difference in amplitude. I think it is better to either describe the two characteristics at the moment or just mention that they have different characteristics and then detail them in the following sentences.

We agree. We have changed the sentence to:

"*The peaks and troughs of the oscillations occur simultaneously upslope and in the valley bottom (piezometer at the weather station), but the peaks upslope are much sharper and have higher  amplitudes (Fig. 7)."* (line 181-182)

L168. Perhaps a more technical term than "the big picture" can be used... Such as "the typical/mean annual cycle".

Yes, we agree, we have replaced the words "the big picture" (line 190).

L252-253. Do you define peak duration as the time the groundwater was less than 50 cm from the ground? I think it is important to clarify.

Yes, we agree – this is important to clarify. We have inserted "when groundwater was less than 50 cm from the surface" in line 278-279 to clarify how we define the duration of the peak.

L261-263. It may be helpful for the reader to indicate the date May 26 in Fig. 6.

Yes, good idea. We have indicated May 26 in our revised Fig. 6.

Figure 1. Snow avalanches are not landslides and it is better to use the full terms. . . Landslide instead of slide and rockfall instead of rock.

Yes, we agree. We have changed the legend in Fig. 1.

Figure 1. A minor issue in the legend: the precipitation is a continuous number. This should also be reflected in the legend. Better write: 0-30; >30-60; >60-90;. . . etc.

Yes, we agree, we have changed this in the revised Fig. 1.

Caption Figure 1. "114 debris flows and slides".

Yes, we agree. We have changed from "114 debris flows, slides,…." to "114 debris flows and slides" – as suggested (line 57).

Figure 2. To respect the structure: "Photo: Jan Helge Aalbu on 16 November 2013."

Yes, we have changed the structure (line 63) and included a new Table (Table 1) with some more information about the landslides in the photos in Fig. 2.

Figure 4. Try to improve the sentence so as not to repeat "aerial photo". One possibility is: "Map and aerial photo from 2018 of the site near Anestølen. A. The contour interval is 100 m. B. The eastern slope is prone to. . ."

Good idea. We have replaced as suggested (Line 155-156).

Figure 5B. It is not necessary to clarify that the rocks are solid.

Agree – we have changed from "Solid rocks" to "Boulders" in Figure 5B.

Response from the authors (in blue) to the comments by anonymous referee (in black), updated 9. April 2021.

[Line numbers in blue refers to the new version of the manuscript: Bondevik_Sorteberg_Groundwater fluctuations_revised_marked_up.docx]

GENERAL COMMENTS

The manuscript provides continuous data about precipitation, air and groundwater temperature, snow depth, pore-water pressure, monitored on a site in Western Norway that in November 2013 was involved in a weather-induced debris flow after a storm that caused about 142 landslides and 7 snow avalanches in the same region. The reported data allow, in particular, to compare the weather and piezometric conditions responsible of the debris flow with those occurred in the past not able to induce any failure. The paper is well written and contains very accurate figures.

We are glad you think the paper is well written and has accurate figures. We appreciate all your suggestions that will improve our paper. Below we comment on each of these and show how we have revised our manuscript.

The availability of information exactly during the landslide event represents undoubtedly a valuable aspect not to be overlooked. However, as also correctly recognized in the text by the Authors, two evident limitations exist: i) the pore-water pressure data are measured only by one piezometer (located upslope); ii) information about the properties of the involved soils are absent. Of course, the first aspect, that prevents to model the piezometric regime along the slope, can not be solved. On the contrary, I hope that some data about the physical, hydraulic and mechanical soil properties should be added because a full comprehension of the landslide is very hard without them. In particular, the absence of information regarding the shear strength parameters makes impossible analyzing the slope stability conditions. Some specific suggestions, aimed to improve the quality of the manuscript, are reported in the following section.

About the properties of the involved soils:

Ideally, it would be very nice to present the properties of the soils on the slope in a table. However, we do not have such data, such data are not available from other studies of the area, and the material on the slope varies a lot, both laterally and with depth, that makes it difficult. In the paper we present the information we have about the properties of the slope deposits that allow the reader to have some understanding of the slope and the soil. Below is line 93-95 in the method chapter:

*"The lower part of the slope is covered by slide deposits and till and the average slope angle is 25 °– 26 °. The sediment-covered slope tapers off upwards into steeper and exposed bedrock and cliffs (Fig. 5A). From the outcrops of the slide scar of the 2007 slide event (Fig. 4B), we found that the thickness of the deposits on the slope varies, probably between 2 and 5 m.*

And also in line 103-104:

*We drilled through boulders and relatively firm deposits down to 2.4 m below the surface using a hammer drill powered by compressed air (Fig. 5B).*

In the caption to Fig. 5B (line 165-169) we say:

*"Average steepness of the sediment-covered slope is 25°–26°. The stippled line indicates where we believe the boundary between till and bedrock is located, based on observations from the 2007 slide scar (Fig. 4B). B. The mini-diver (sensor) is suspended from a wire in the piezometer and anchored 1.64 m below the surface. The top 60 cm of deposits consists of organic soil and weathered till, with firm till below that. We drilled through boulders at depths of ca. 1 m and 1.5 m."*

And, in the discussion chapter we say (line 250-252):

*"The slope is covered by till and colluvium, deposits that vary widely in composition and grain size. Thus, observations from one borehole may not be representative of other areas on the same slope."*

In order to provide accurate data about the soil properties, especially data for cohesion and friction angle (the shear strength parameters), and data of the hydraulic conductivity etc., we would need to do a new study of the slope. Because of the heterogeneity of the soil, we would need information from several locations along the slope and also from different depths. On an ideal slope with homogeneous soils such data would be very useful, and possible also available. In addition, in this paper, we do not perform any calculations or simulations of stability or of the slide movement. We believe that the information given is sufficient for the reader to get the information of the soils on the slope to understand the work and data we present about the groundwater fluctuations during the debris flow event.

SPECIFIC COMMENTS

Line 30. The availability of real-time water level data during rapid landslides are effectively rare, but, on the contrary, many papers provide the pore pressure in slopes involved in active slow landslides, cyclically reactivated by seasonal weather events. Therefore, the sentence "rare because it is difficult to predict which slope will fail" should be replaced by "rarely provided during rapid landslide events".

We agree, thanks for this suggestion, we have replaced as suggested (line 30).

Lines 39-40.

Snow avalanches are not landslides. Therefore, the sentence "Most of the slides were debris slides and flows (114), but rockfalls (28) and snow avalanches (7) also occurred" should be replaced by "Most of them were debris slides and flows (114), but rockfalls (28) also occurred. Some snow avalanches (7) were observed too".

We agree. We have changed "slides" with "mass movements". The sentence have been changed to (line 42-43):

*"Most of the  mass movements were debris slides and flows (114), but rockfalls (28) and snow avalanches (7) also occurred (Fig. 1).*

Figure 1. The term "slides" in the legend can not be used to indicate at the same time the three types of phenomena. It should be replaced by a term like "Events". Moreover, I suggest to indicate them according to the following order: i) Debris flow and slide; ii) Rockfall; iii) Snow avalanche.

Yes, we agree and have changed it as you suggested here, both in figure caption (line 56-57) and on the revised figure (legend).

Caption Figure 2.

I suggest to simplify it, inserting in a table (to be cited in the text) all the provided information regarding the three shown landslides: date and hour of the occurring events, landslide length, upslope and downslope altitudes, mean slope inclination, range of thickness, etc.

It is a good idea. We have provided a Table (Table 1, Line 70-75) with this information and simplified the caption to Fig. 2 (line 62-70).

Section 2.

This section should contain a table reporting the available information (eventually deriving them by other papers) about the mean values of physical, hydraulic and mechanical properties of the involved soils: grain size, in-situ porosity and degree of saturation, unit weight, hydraulic conductivity, strength parameters. Such values are very important to allow a full understanding of the infiltration and seepage processes and, as a consequence, of the induced landslide mechanism.

See comment above under the heading "About the properties of the involved soils."

Line 90.

Indicate at which altitude and distance from the toe of the landslide the piezometer has been installed.

OK, we have added a sentence that gives that information in line 104-105.

Line 125.

Indicate the total length of the debris flow.

OK, added a sentence that gives the length of the debris flow (line 145). This length can also be found in the new Table 1.

Figure 5A. Clarify which "Distance" is reported in the X-axis. Is it the distance from the toe of the landslide ?

No, it is the distance from the road on the flat valley bottom and upslope to the borehole, at 90° to the slope, and parallel to the slide scar of the 2007 landslide. We have indicated the location of the profile on the aerial photo in Fig. 4B, as a stippled line, that will clarify better the location of the profile.

Figure 6. According to the results, the influence of the snow cover melting on the water level is particularly important. Therefore, I suggest to insert in this figure the data about the snow depth (shown only by the supplementary Figure S9) and about the air temperature (partially shown in Figure 8).

We have moved the snow cover figure from the supplements and included it into the revised Figure 6 (lower diagram). We also tried to include the air temperature graph, but that made figure 6 too messy. We think the snow cover figure works nicely in Fig. 6 and we think it is fine to keep the air temperature graph for the whole year in the supplements. However, air temperatures for the days around the slide event are shown in Fig. 8.

Caption Figure 7. Indicate at which depth and altitude the piezometer of the weather station has been installed.

The altitude of the weather station (and piezometer) is given in the method section of the paper.

Figure 8. Due to the important role of the snow melting, I suggest to insert in this figure the data about the snow depth (shown by the supplementary Figure S9).

Melting of snow has an important role, but we are uncertain that the measurements from the snow pillow at the day the slide happened, with so low values, can be fully trusted. The snow pillow show 0.020 m of water equivalent before the event, and during the event rises to 0.024 m. This value is a combination of snow that was on the ground before the event, new snow that fell during the first phase of the event and water from the change in precipitation to rain. Since the snow pillow cannot be trusted with such low values, we have decided not to include it in such details as the data in Fig. 8 is presented.

Line 222.

The second important weak point of the manuscript regards the absolute absence of information about the soil properties. As already suggested, I hope that you are able to provide them. For instance, some information about the strength parameters could help (at least) estimating the slope stability conditions.

Again, see comment above under the heading "Information about the properties of the involved soils"

Lines 265-266.

Differently from what observed in November 2013, the piezometric peaks monitored in April and May 2013 were caused only by rainwater infiltration (and not by snow melting). Why do you consider such evidence so relevant to not induce sliding ? The corresponding measured peaks of 33 cm (measured in April) and 28 cm (measured in May) below the ground surface are very close to the critical estimated value of 30 cm in November 2013 (such value was extrapolated from the groundwater level curve measured from 19:00 and 23:00, as clarified in Lines 237-238). As a consequence, being the local shear strength approximatively the same at the onset of the three attained maximum water levels, the corresponding local slope stability conditions should be essentially the same too. Unfortunately, the availability of only one piezometer does not allow to make a reliable evaluation of the general slope stability conditions, therefore your consideration seems rather rash. Please make some comments.

We agree; the very high peaks in May and April is essential the same values as the critical peak in November 2013, but we now think, thanks to this review comment that encouraged more thinking about this, that the soil on the slope was probably (partly) frozen in April and May, that could have prevented effective drainage of groundwater downslope. The thawing of the frozen soil, melting of snow on the ground and the rain episodes caused the high piezometric peaks. In spite of the high pore pressures in the borehole, the slope was maybe not unstable, because part of the soil on the slope was still frozen, and a frozen pore space would give higher shear strength. The air temperatures in April and May was low, most of the nights were below freezing until May 5, that would prevent the thawing of the ground and refreezing of the surface water during nights.

We have rewritten this paragraph (line 286-293) and enlightened this possibility of frozen soils in April and May.

Lines 276-279.

The emphasis of provided considerations is rather strange. It's well known that the initial conditions are crucial to determine the weather-induced effects. Once given an initial monitored piezometric value, the main challenge should be, of course, associating a landslide hazard to a forecasted weather event. At the same time, associating a very low landslide hazard to severe weather event if the initial measured groundwater level is located below a "safe" value should be also very useful for the implementation of an early warning system. I encourage the Authors to make some comments about this topic.

Yes, we have added a new paragraph (line 320-327) about the early warning system in Norway and suggested here that there might be a good idea to include piezometers from slopes susceptible to debris flows to the warning system.

TECHNICAL CORRECTIONS

Title. Is the hyphen "-", between "Norway" and "triggered", necessary ?

It is not necessary for us to use "–", a comma instead would also work. However, using "–" (em dashes) to replace commas makes the reader focus a bit more on this information that is set inside the em dashes "–". We have thus kept the em dashes in the title.

Caption Figure 1. "114 debris flows, slides" should be replaced by "114 debris flows and slides"

Yes, we have changed accordingly.

Line 109.

The word "from" at the end of the line should be replaced by "carried out".

? We do not understand this comment …."carried out"… and have not replaced the word "from" in our revised text.

Line 163.

The sentence "has also been" should be replaced by "already".

OK, replaced "has also been" with "already".

[revised manuscript text omitted]